# Clinical Practice Evolvement for Post-Operative Prostate Cancer Radiotherapy—Part 2: Feasibility of Margin Reduction for Fractionated Radiation Treatment with Advanced Image Guidance

**DOI:** 10.3390/cancers15010040

**Published:** 2022-12-21

**Authors:** Brady S. Laughlin, Nathan Y. Yu, Stephanie Lo, Jingwei Duan, Zachary Welchel, Katie Tinnon, Mason Beckett, Steven E. Schild, William W. Wong, Sameer R. Keole, Jean-Claude M. Rwigema, Carlos E. Vargas, Yi Rong

**Affiliations:** 1Department of Radiation Oncology, Mayo Clinic, Phoenix, AZ 85259, USA; 2Department of Radiation Oncology, University of Kentucky, Lexington, KY 40506, USA; 3Department of Nuclear and Radiological Engineering, Georgia Institute of Technology, Atlanta, GA 30332, USA

**Keywords:** post prostatectomy radiotherapy, advanced image guidance, iCBCT, PTV margins

## Abstract

**Simple Summary:**

While the definition of the prostate bed clinical target volume (CTV) has been proposed in consensus guidelines, various PTV expansions have been proposed. Depending on the institution, isotropic or non-isotropic expansions on the prostate bed CTV may be considered. This is a preliminary study evaluating PTV margin reduction accounting for inter-fractional uncertainties using high quality iCBCT image guidance. This analysis revealed that 2 mm and 4 mm margins may be considered for margin expansion with and without an endorectal balloon.

**Abstract:**

Purpose: Planning target volume (PTV) expansion for post-prostatectomy radiotherapy is typically ≥5 mm. Recent clinical trials have proved the feasibility of a reduced margin of 2–3 mm for treatments on MRI-linac. We aim to study the minimum PTV margin needed using iterative cone-beam CT (iCBCT) as image guidance on conventional linacs. Materials/Methods: Fourteen patients who received post-prostatectomy irradiation (8 with an endorectal balloon and 6 without a balloon) were included in this study. Treatment was delivered with volumetric modulated radiation therapy (VMAT). Fractional dose delivery was evaluated in 165 treatment fractions. The bladder, rectal wall, femoral heads, and prostate bed clinical tumor volume (CTV) were contoured and verified on daily iCBCT. PTV margins (0 mm, 2 mm, and 4 mm) were evaluated on daily iCBCT. CTV coverage and OAR dose parameters were assessed with each PTV margin. Results: CTV D100% was underdosed with a 0 mm margin in 32% of fractions in comparison with 2 mm (6%) and 4 mm (6%) PTV margin (*p* ≤ 0.001). CTV D95% > 95% was met in 93–94% fractions for all PTV expansions. CTV D95% > 95% was achieved in more patients with an endorectal balloon than those without: 0 mm—90/91 (99%) vs. 63/74 (85%); 2 mm—90/91 (99%) vs. 65/75 (87%); 4 mm—90/90 (100%) vs. 63/73 (86%). There was no difference in absolute median change in CTV D95% (0.32%) for 0-, 2-, and 4 mm margins. The maximum dose remained under 108% for 100% (0 mm), 97% (2 mm), and 98% (4 mm) of images. Rectal wall maximum dose remained under 108% for 100% (0 mm), 100% (2 mm), and 98% (4 mm) of images. Conclusions: With high-quality iCBCT image guidance, PTV margin accounting for inter-fractional uncertainties can be safely reduced for post-prostatectomy radiotherapy. For fractionated radiotherapy, an isotropic expansion of 2 mm and 4 mm may be considered for margin expansion with and without the endorectal balloon. Future application for margin reduction needs to be further evaluated and considered with the advent of shorter post-prostatectomy radiation courses.

## 1. Introduction

Radiation to the prostate bed is recommended for prostate cancer patients with high-risk pathologic features or who develop biochemical failure following radical prostatectomy [1]. While the definition of the prostate bed clinical target volume (CTV) has been proposed in consensus guidelines [2,3,4,5], there is no clarity regarding the proper planning tumor volume (PTV) expansion margin. This may be attributable to considerable variation in target location and deformation of the prostate bed caused by the interfraction and intrafraction changes in the bladder and rectum volume compared to the intact prostate [6]. Dose coverage of the prostate bed and organs-at-risk (OAR) also becomes more complicated secondary to such variation in CTV position and volume. Accurate delivery of post-prostatectomy radiotherapy would require proper evaluation of prostate bed CTV in daily positioning and appropriate margins [7]. High-quality image guidance radiotherapy (IGRT) practices must be set up to ensure accurate delivery of planned radiation treatment. PTV margins should be determined based on the achievable accuracy and consistency of IGRT and operators’ training competency [8].

Accurate IGRT techniques with volumetric imaging, i.e., cone beam CT (CBCT), are imperative for adapting PTV margins to ensure correct dose delivery. For post-prostatectomy radiation, there has not been a consensus standard PTV expansion. Typically, these margins range from 3 mm to 10 mm [9]. The literature has not provided consistent recommendations. Huang et al. showed that a reduced PTV margin of 3–5 mm for prostatectomy radiotherapy using daily CBCT could achieve sufficient target dose coverage and sparing of surrounding OARs [7]. On the contrary, an analysis of 477 CBCT in 15 patients showed a posterior geometric miss in 10% of fractions using a PTV margin of 5 mm posterior and 10 mm in all other directions [10]. Furthermore, a retrospective analysis of CBCT setup in a phase II trial for patients receiving stereotactic body radiotherapy (SBRT) in the post-prostatectomy setting demonstrated that although CTV inter-fractional change was small, CTV V95% > 93% was met in 70% of cases with a 5 mm PTV margin [11]. This inconsistency might be multi-fold, including but not limited to using an endorectal balloon, variation in bladder volume, daily imaging quality, image registration accuracy, and treatment therapists’ expertise. 

With technological advancements in image guidance, smaller margins may be considered for post-prostatectomy radiation. Recent clinical trials have investigated hypofractionation and SBRT in the post-prostatectomy setting [11,12,13,14]. In studies implementing SBRT using an MRI linac, it was shown that reduced margins may achieve a high delivery dose per fraction to the target while adequately sparing OAR [11,12]. However, MRI linacs are not widely available. For centers with conventional linacs, iterative reconstructed cone beam CT (iCBCT) provides an enhanced image reconstruction technique with improved soft tissue contrast and accurate HU values compared to a standard CBCT [15,16]. Therefore, implementing iCBCT as image guidance may allow better image guidance with reduced margins to account for target setup uncertainty and minimize inter-fractional motion. 

Most margin studies were performed based on the assessment of residual shifts after IGRT or corrected shifts between multiple imaging throughout the treatment, and the overall margins were calculated using one of the margin calculation equations [17,18]. This approach is straightforward but with multiple assumptions and simplifications, which may cause inconsistent margin conclusions, especially for those anatomical sites without a solid or visible target, including post-prostatectomy cases. Our study was designed to directly compute dose on daily iCBCT images and evaluate delivery accuracy with different PTV margins. Similar studies were conducted years ago but used traditional Feldkamp-Davis-Kress (FDK) filter-back projection algorithm-based CBCT images, which had poor soft tissue contrast due to scatter-contamination [10]. Herein, we aim to study if iCBCTs allow for reduced margin expansions accounting for inter-fractional motion while maintaining sufficient target coverage. The impact of 3 isotropic PTV margin expansions (0-, 2-, and 4 mm) on dose delivery to the CTVs and adjacent OARs was assessed using 165 fractions of daily iCBCT images. 

## 2. Materials and Methods

### 2.1. Patient Characteristics, Simulation, and Planning

Institutional Review Board approval was obtained for this study. One-hundred and sixty-five iCBCT (every 3 fractions) from 14 patients who underwent salvage PPRT were analyzed. The most common dose/fractionation was 66 Gy in 33 fractions, delivered in 10 patients (71.4%). Other dose/fractionation schemes, including 52.5 Gy/20 fractions and 70.2 Gy/39 fractions, were delivered to 2 patients each (17.4%). 

All patients were instructed to empty the bladder and drink 16 oz of water 45 min before simulation and daily treatment. Based on physician preference, an endorectal balloon was used for 8/14 (57%) patients.

Prostate bed CTV was contoured per the Faculty of Radiation Oncology Genito-Urinary Group (FROGG) consensus guideline. An MRI was utilized to contour the apex of the prostate to the plane where the puborectalis is at the level of the urethra. The ipsilateral seminal vesicle bed was included if seminal vesical invasion was present. The retropubic space was included for the initial inferior half of the pubic bone. OARs, including bladder, rectum, femoral heads, and small and large bowel, were contoured for treatment planning and optimization. The rectal wall was defined as the outermost 3 mm of the rectum. Volumetric modulated arc therapy (VMAT) was planned for all patients. For making consistent plans with multiple PTV margins, an inverse planning optimizer, RapidPlan™ (Varian Medical Systems, Palo Alto, CA, USA), was used with minimal human intervention, adapting to PTV margins using the same prostate-bed model: bladder V65% < 60%, bladder D0.03cc < 108%, rectal wall D0.03 < 108%, CTV D95% > 95% of the prescription dose.

### 2.2. Data Acquisition and Analysis

Clinically acceptable plans with 3 isotropic PTV margin expansions (0-, 2-, and 4 mm) were generated using the same RapidPlan model used for each patient’s original clinical plan, and similar plan quality was achieved in plans with different margins without human intervention. Figure 1a–c shows one example with three plans of 0-, 2-, and 4 mm PTV margins in the axial, coronal, and sagittal views for a patient with an endorectal balloon. Similarly, Figure 1d–f show plans for patients without an endorectal balloon. Treatment image guidance protocol using iCBCTs was strictly followed, as discussed in Part 1 [19]. The couch has six degrees of freedom, which allows translational and rotational couch correction after daily imaging matching. Three PTV margin VMAT plans were copied from the original CT of each patient to the corresponding daily iCBCTs, which were calibrated and proved to be within 2% dose calculation accuracy [16]. Daily image registration isocenter was used as the beam isocenter for plan re-calculation, which simulated the dose that could have been delivered for the day of treatment if the tested plan was used. For each iCBCT, a radiation oncologist copied and approved prostate bed CTV, and organs-at-risk were contoured. For every iCBCT, little to no deformation was appreciated and may only be accounted for by intraobserver variation. The following dose volume points were accessed for daily delivery using 3 different PTV margins: Bladder V65% < 60%, bladder D0.03cc < 108%, rectal wall D0.03 < 108%, CTV D90%, D95%, D98%, and D100%. Additional dose volume goals were assessed for plan quality: Bladder: mean and median dose, V75Gy < 25%, V70Gy < 35%, V65Gy < 50%, V40Gy < 70%, V65% < 60%, and D0.03cc < 108%. Rectum: mean and median dose, V75Gy < 15%, V70Gy < 25%, V65Gy < 35%, and V40Gy < 55%. Dose-volume histogram data was extracted from the Eclipse treatment planning system (Varian Medical Systems, Palo Alto, CA, USA) and analyzed using MATLAB Version R2021b (MathWorks, Inc., Natick, MA, USA). The one-way ANOVA test and Tukey post hoc test were used to determine any statistically significant differences between the means of 0-, 2-, and 4 mm isotropic PTV expansions using SPSS (SPSS Inc., Chicago, IL, USA). Violin plots demonstrating the distribution and density of target coverage and dose to OARS were generated using Python (Python Software Foundation, Wilmington, DE, USA).

## 3. Results

The interfraction absolute median change in CTV V95% was 0.32% for all margin expansions. The CTV DVH parameters of D90%, D95%, D98%, and D100% for patients treated with and without an endorectal balloon are demonstrated in a violin plot (Figure 2). CTV D95% was greater than 95% for 153 (93%), 155 (94%), and 155 (94%) fractions for 0-, 2-, and 4 mm margins, respectively. CTV D95% > 95% was achieved in more patients with an endorectal balloon than those without: 0 mm—90/91 (99%) vs. 63/74 (85%); 2 mm—90/91 (99%) vs. 65/75 (87%); 4 mm—90/90 (100%) vs. 63/73 (86%). CTV D95% > 90% for all fractions for 2- and 4 mm plans and all except one for 0 mm. There was significant under-coverage of the CTV D100% with the 0 mm margin (52/165, 32%) in comparison with the 2 mm (10/165, 6%) and 4 mm (10/165, 6%) PTV margin (*p* ≤ 0.001) in patients with and without an endorectal balloon. There was no difference in CTV D100% between the 2 mm and 4 mm plans (*p* > 0.05). For patients with an endorectal balloon, there was no difference in CTV D98%, D95%, and D90% between the 0 mm, 2 mm, and 4 mm expansions (*p* > 0.05). For patients without an endorectal balloon, there were significant under-coverage of CTV D98% (*p* ≤ 0.05) between 0- and 4 mm margins. There was no difference in coverage for CTV D95% and D90% for 0-, 2-, and 4 mm expansions in patients without an endorectal balloon (*p* > 0.05). 

Figure 3 demonstrates V65% relative to daily bladder volume as a function of daily bladder volume (Figure 3a–c) for uniform PTV expansions of 0-, 2-, and 4 mm. For each margin expansion, it is demonstrated that the bladder V65% < 60% is met in most cases as bladder volume approaches 100–150 cc. Figure 3a–c demonstrates that the V65 of the bladder decreases as the daily volume of the bladder increases. The bladder V65% < 60% constraint was not met in 4 (2%), 5 (3%), and 6 (4%) fractions for 0-, 2-, and 4 mm margins, respectively. The median bladder V65% was 17.6%, 18.6%, and 19.7% for 0-, 2-, and 4 mm margins, respectively. The interfraction median absolute change in bladder V65% was 4–4.8% for the different margin expansions. The maximum rectal wall (3 mm) dose is demonstrated as a function of daily bladder volume for various target margins in Figure 3d–f. All fractions for the 0- and 2 mm margin plans resulted in a maximum rectal wall dose that fell under the 108% limit. In plans with a 4 mm PTV margin, three (3%) fractions resulted in maximum rectal wall doses that exceeded 108%, all in patients with an endorectal balloon. The median rectum D0.03cc were 102.2%, 103.7%, and 104.3% for 0-, 2-, and 4 mm in patients with an endorectal balloon, and 101.9%, 103.0%, and 103.9% for 0-, 2-, and 4 mm in patients without an endorectal balloon. The maximum bladder dose (D0.03 cc) is provided in Figure 3 as a function of normalized daily bladder volume (Figure 3g–i). For the fractions calculated with 0 mm plans, all maximum doses fell under the institutionally established limit of 108%. However, for the 2 mm plans, 9 fractions featured a maximum dose exceeding 108%. For the 4 mm plans, the 108% maximum bladder dose limit was exceeded for 5 fractions. 

Rectum DVH parameters (V40Gy, V65Gy, V70Gy, and V75Gy) are demonstrated in violin plots in Figure 4. The violin plots demonstrate the distribution of rectal dose relative to rectum volume. In patients with an endorectal balloon, there were significant differences in rectal V65Gy, V70Gy, and V75Gy between 0 mm and 2 mm (*p* ≤ 0.05), 0- and 4 mm plans (*p* ≤ 0.001). For rectal V75Gy, there was a significant difference between 2 mm and 4 mm plans (*p* ≤ 0.05). There were significant differences in rectal V65Gy, V70Gy, and V75Gy between 0 mm and 2 mm (*p* ≤ 0.05), 0- and 4 mm plans (*p* ≤ 0.001), and 2- and 4 mm plans (*p* ≤ 0.05) in patients without an endorectal balloon. Figure 5 demonstrates distributions of various bladder dose-volume points. With the primary bladder constraint V65 < 60% There was no difference in bladder dose between 0-, 2-, and 4 mm plans. The median bladder D0.03cc was 104.1%, 104.1%, and 104.9% for 0-, 2-, and 4 mm, respectively.

## 4. Discussion

We demonstrated that a 0 mm PTV margin with an endorectal balloon and a 2 mm PTV margin without an endorectal balloon is feasible, accounting for inter-fractional uncertainty for post-prostatectomy RT, with adequate target coverage, reduced maximum rectum dose, and no change in bladder dose. As demonstrated in Part 1 [19], there was little impact on variation in bladder filling or rectum on deformation of the CTV on iCBCTs. CTV coverage of D100% ≥ 95% was achieved using a 0 mm margin in all patients with an endorectal balloon. The same CTV coverage was achieved using a 2- or 4 mm margin in all patients without an endorectal balloon. This is the first study to evaluate inter-fractional uncertainty improvement using iCBCT images and one of the few to evaluate the impact of PTV margin on target coverage and OAR DVH parameters by calculating dose directly on daily CBCTs. 

Various groups have assessed the impact of different PTV margins on patients undergoing post-prostatectomy RT [10,20,21]. Gill et al. applied two PTV margins, 10 mm isometrically (PTV10) and 5 mm posteriorly and 10 mm (PTV5) in all other directions to the planning CTV, and found 46 posterior geographic misses in the PTV5 group in comparison with 26 posterior geographic misses in the PTV10 group [10]. Bell and colleagues evaluated 6 combinations of anisotropic PTV margins to determine rates of geographic miss and doses to the bladder and rectum [20]. The optimal PTV expansion was 5 mm in all directions except 10 mm in the anterior-posterior direction for the upper prostate bed, which resulted in both low rates of geographic miss and reduced dose to the bladder and rectum [20]. However, Both et al. demonstrated that a 3 mm margin was sufficient for adequate coverage of intra-fractional prostate motion [22]. Our study demonstrated that 2 mm inter-fractional margins should maintain sufficient CTV coverage with no significant change in meeting dose constraints for the bladder and rectum. The success of further reduction of PTV margin in our study is likely attributable to modern iCBCT and well-trained therapists to ensure minimal inter-fractional uncertainties. 

Advances in IGRT have improved accuracy for both intact prostate and post-prostatectomy patients. Preliminary analysis of a phase III trial evaluating MRI-guided SBRT vs. CBCT-guided SBRT for intact prostate cancer patients demonstrated a reduction in acute grade > 2 GU toxicity [23]. The PTV margin was 2 mm in the MRI arm vs. 4 mm in the CBCT arm due to the limited contrast visualization in organ interfaces using conventional FDK-based CBCT. For post-prostatectomy patients, FDK-based CBCT is inadequate for ensuring CTV coverage when isotropic 5 mm expansion was used, with only 72% (13/18) patients achieving CTV V95% greater than 93% [11]. The same group evaluated the advancement in daily MRI guidance for prostate cancer patients receiving post-prostatectomy SBRT [24]. MRI guidance was essential in mitigating variations in volumetric changes in 78.2% of fractions due to exceeding OAR constraints and target under-coverage [24]. Our study suggests we can achieve a similar outcome with a reduced PTV margin for CBCT-guided RT using advanced iCBCT IGRT. 

The interfraction and intrafraction components of prostate bed motion are essential in evaluating the accurate delivery of post-prostatectomy radiation. With adequate target coverage defined as CTV D_min_ > 95%, Zhu et al. demonstrated that rotational motion in 5 of 16 patients was associated with target under-coverage [25]. Bell et al. measured intra-fraction displacement in 46 patients (392 post-treatment CBCT) receiving post-prostatectomy radiation treated with anisotropic margins and daily soft tissue matching [26]. Although intrafraction displacement was small (0.7–1.3 mm) in all directions, it was a significant contributor to an 8.4% marginal miss rate despite soft tissue matching and anisotropic margins (5 mm in all directions of lower and upper prostate bed except 10 mm anteriorly and posterior in upper prostate bed) [26]. Both et al. evaluated intrafraction prostate motion in 787 sessions for 24 patients with endorectal balloons, utilizing implanted electromagnetic transponders, finding that a 3 mm internal margin can account for 95% of intrafraction motion [22]. The traditional method of calculating optimal PTV margins is to assess systemic and random components of interfraction and intrafraction prostate motion errors and plug them into the Van Herk equation (2.5 × Σ + 0.7 × σ) [18]. However, considering the abovementioned literature on intra-fractional motion, our study design concludes that a margin of 2–3 mm and 4–5 mm might be sufficient in ensuring CTV coverage with and without an endorectal balloon, respectively. 

This study has several limitations. Although PTV margins were evaluated on 165 daily fractions of iCBCT, this only corresponded to 14 patients. However, given the similar anatomy of prostate bed patients, we believe we are accounting for bladder and rectum variation, despite the small cohort of patients. According to Yoon and colleagues, there was minimal deformation in CTV volume in patients receiving post-prostatectomy SBRT [11]. In a follow-up study involving MR guidance published by Cao et al., there was slight variation in shape change of the CTV with dice similarity coefficient of 0.83 [24]. Volume changes reported in this study cannot be directly applied to our study as changes depend on the specific study group and treatment duration. As more recent trials have introduced shorter courses of post-prostatectomy radiotherapy, CTV volume changes are likely to be smaller than changes previously reported. Additionally, the development of prostate bed CTV guidelines has been defined by recurrence patterns established after landmark trials. Prostate bed CTV defined on the older version of CBCTs is associated with high user variation, and might not be accurate given poor image quality. As shown in Part I, iCBCT with improved image quality provides higher confidence in determining daily prostate bed CTV. We have confirmed no deformation of the prostate bed CTV in 165 datapoints for the 14 patients we studied. Nevertheless, this is a limitation of our study, which is that the conclusion applies to common anatomy without large size variations. We are expected to see higher dosimetric variation for those patients with potentially larger anatomy variations, which can then be discovered and monitored by high quality iCBCT images. The PTV margin for those scenarios should be subject to physician discretion. Additionally, we recognize the lack of evaluation of non-isotropic margins as a limitation of our study. [26] However, there may be heterogeneity in the field with application of isotropic or non-isotropic expansions. It is our clinical practice to use isotropic expansion. 

Furthermore, the intra-fractional motion of prostate-bed CTV was considered based on literature instead of being evaluated. However, we felt that this value for our patients should not deviate from the literature if the treatment delivery time and patient setup were kept similar to that reported in the literature [27]. Finally, the study cohort for patients with and without ERB is relatively limited. However, based on the consistency of the prostate anatomy and planning dose distribution, we felt the conclusion was valid and justified with the studied cohort, and adding more data points will not alter the study conclusion. 

## 5. Conclusions

In this study, we demonstrated that image guidance with iCBCT may allow for reduced PTV margin in PPRT. Clinical trials for PPRT can consider reduced margins if using high-quality iCBCT for image guidance.

## Figures and Tables

**Figure 1 cancers-15-00040-f001:**
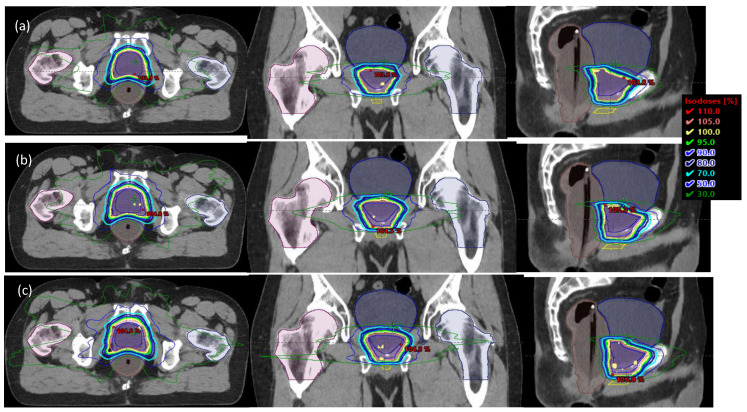
Three plans of 0-, 2-, and 4 mm PTV margins in the axial, coronal, and sagittal views for a patient with (**a**–**c**) and without (**d**–**f**) an endorectal balloon.

**Figure 2 cancers-15-00040-f002:**
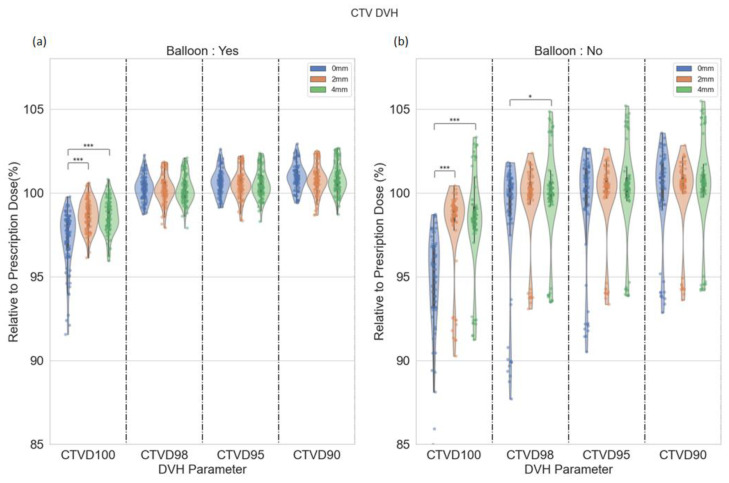
Violin plot for CTV DVH parameters for patients with (**a**) and without (**b**) a endorectal balloon. The annotation ‘*’ denotes statistical significance with *p* ≤ 0.05, and ‘***’ denotes *p* ≤ 0.001.

**Figure 3 cancers-15-00040-f003:**
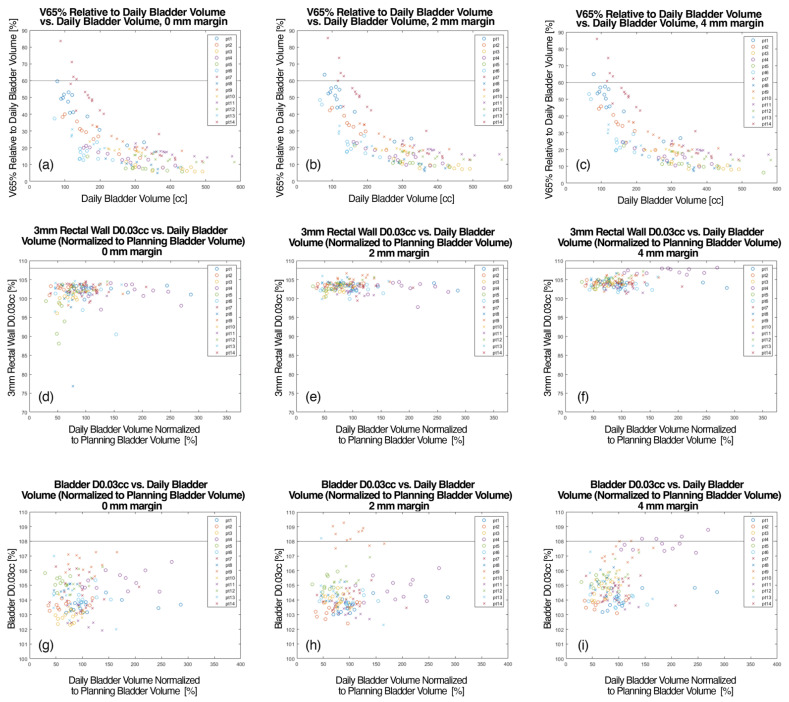
(**a**–**c**) Daily bladder volume that passed the V65% < 60% constraint versus daily bladder volume for different PTV margins (0-, 2-, and 4 mm). (**d**–**f**) Maximum rectal wall (3 mm) dose as a function of daily bladder volume for various target margins. (**g**–**i**) Maximum bladder dose (0.03 mL) versus daily bladder volume for different PTV margins (0-, 2-, and 4 mm).

**Figure 4 cancers-15-00040-f004:**
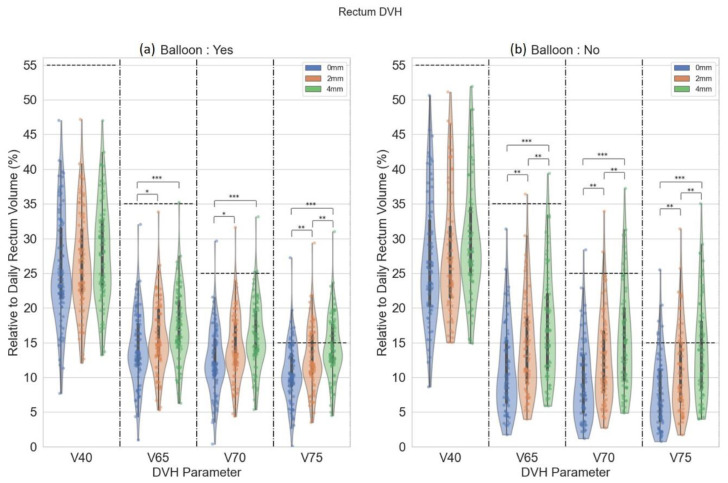
Violin plot for rectum DVH parameters for patients with (a) and without (b) an endorectal balloon. The parallel dash lines indicate the dose constraints. The annotation ‘*’ denotes statistical significance with *p* ≤ 0.05, ‘**’ denotes *p* ≤ 0.01, and ‘***’ denotes *p* ≤ 0.001.

**Figure 5 cancers-15-00040-f005:**
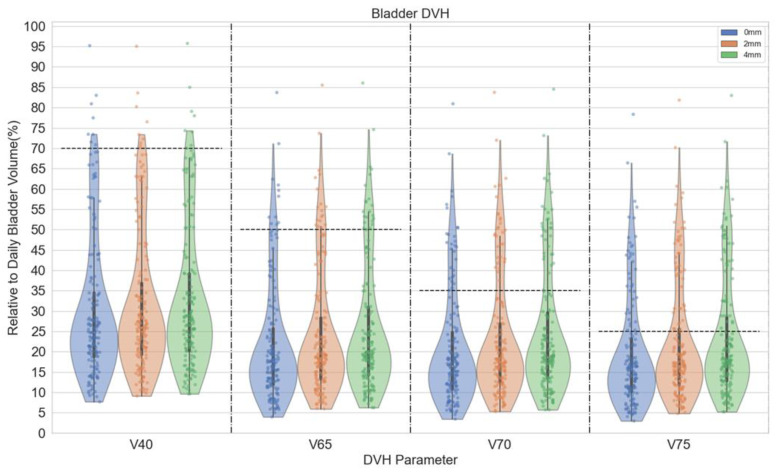
Violin plot for bladder DVH parameters for all patients. The parallel dash lines indicate the dose constraints. These DVH parameters were listed to show plan quality change. Dashed lines were commonly used dose constraints from literature or other practice guidelines.

## Data Availability

Research data are stored in an institutional repository and will be shared upon request to the corresponding author.

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
