# Peer review of "Clinical Practice Evolvement for Post-Operative Prostate Cancer Radiotherapy—Part 2: Feasibility of Margin Reduction for Fractionated Radiation Treatment with Advanced Image Guidance"

_cancers, 2022, doi:10.3390/cancers15010040_

Round 1
Reviewer 1 Report
I enjoyed reading your report. Please consider checking some details:
Abbreviation: in M and M you use PPRT but in conclusion you use post-prostatectomy RT. Please present the acronymous.
Punctuation
In some part you write 10mm or 10 mm. Please unify.
Please leave a space between the final Word of the sentence and [] throughout the manuscript. E.g. [13,14]
References
Ref 9 the title is [] and ref 16 the doi is duplicated
Author Response
Reviewer # 1
Abbreviation: in M and M you use PPRT but in conclusion you use post-prostatectomy RT. Please present the acronymous.
Thank you for this comment. We have adjusted the conclusión to say the acronym.
Punctuation
Thank you for suggest to look at punctuation. We have made some changes.
In some part you write 10mm or 10 mm. Please unify.
Thank you for this comment. We have found one instance where it says 10mm and changed it to 10 mm.
Please leave a space between the final Word of the sentence and [] throughout the manuscript. E.g. [13,14]
Thank you for this suggestion. We have adjusted the
References
Ref 9 the title is [] and ref 16 the doi is duplicated
Thank you for this point. We have made the adjustments as suggested.

Reviewer 2 Report
This is a retrospective treatment planning study to show the PTV margin of post-prostatectomy radiotherapy can be reduced without compromising CTV coverage. The manuscript is well written and easy to follow. However, the approach may be too simple. The data in its current form is not convincing enough to support the conclusion. Even though the authors mentioned in the discussion that there is little impact of variation in bladder and rectum filling on the CTV deformation, that is different from what we observe in the daily imaging. The inter-fraction anatomy variation in post-op prostate patients due to bladder and rectum filling can be very significant albeit rigorous bladder and bowel prep instructions. Copying the three PTV-margin plans from the planning CT to the corresponding daily iCBCTs without deformably re-contouring does not account for the CTV variation caused by normal organ anatomy changes. Significant portion of the CTV to PTV margin is to warrant CTV’s dose coverage in those scenarios. In addition, the intra-fraction CTV changes due to bladder filling is not taken into consideration in this study since only pre-treatment iCBCT was acquired and studied. We have to take a leap of faith to believe the CTV does not change during the course of the treatment. There are studies demonstrating bladder volume increases while patient is on the table.
One idea is to change the study design by re-contouring CTV based on daily high quality iCBCT and show reducing PTV margin can potentially compromise the CTV dose coverage?
Figure 1(d) is missing CTV contour?
Author Response
This is a retrospective treatment planning study to show the PTV margin of post-prostatectomy radiotherapy can be reduced without compromising CTV coverage. The manuscript is well written and easy to follow. However, the approach may be too simple. The data in its current form is not convincing enough to support the conclusion. Even though the authors mentioned in the discussion that there is little impact of variation in bladder and rectum filling on the CTV deformation, that is different from what we observe in the daily imaging. The inter-fraction anatomy variation in post-op prostate patients due to bladder and rectum filling can be very significant albeit rigorous bladder and bowel prep instructions. Copying the three PTV-margin plans from the planning CT to the corresponding daily iCBCTs without deformably re-contouring does not account for the CTV variation caused by normal organ anatomy changes. Significant portion of the CTV to PTV margin is to warrant CTV’s dose coverage in those scenarios. In addition, the intra-fraction CTV changes due to bladder filling is not taken into consideration in this study since only pre-treatment iCBCT was acquired and studied. We have to take a leap of faith to believe the CTV does not change during the course of the treatment. There are studies demonstrating bladder volume increases while patient is on the table.
Thank you for making these points. We have made the following addition to our limitations section:
As shown in Part I, iCBCT with improved image quality provides higher confidence in determining daily prostate bed CTV. We have confirmed no deformation of the prostate bed CTV in 165 datapoints for the 14 patients we studied. Nevertheless, this is a limitation of our study, which is that the conclusion applies to common anatomy without large size variations. We are expected to see higher dosimetric variation for those patients with potentially larger anatomy variations, which can then be discovered and monitored by high quality iCBCT images. The PTV margin for those scenarios should be subject to physician discretion. Additionally, we recognize the lack of evaluation of non-isotropic margins as a limitation of our study. [23] However, there may be heterogeneity in the field with application of isotropic or non-isotropic expansions. It is our clinical practice to use isotropic expansion.
- One idea is to change the study design by re-contouring CTV based on daily high quality iCBCT and show reducing PTV margin can potentially compromise the CTV dose coverage?
Response: Thank you for this suggestion. We agree that this approach would certainly reduce some of the limitations of our study. However, this approach introduced several other uncertainties, including inter-observer variations in defining prostate bed CTV and CTV definition accuracy based on iCBCT image quality. At the study design phase, we have weighted pros and cons of different approaches and decided on this current approach, since propagating CTV from the planning CT to the daily iCBCT with user evaluation and confirmation will not introduce additional uncertainties as mentioned above. We did acknowledge limitations of this study design in the last paragraph of the discussion.
- Figure 1(d) is missing CTV contour?
Response: The CTV contour is shown as the blue shaded area encompassed by the 100% yellow isodose line on Figure 1 (d), (e), and (f).

Reviewer 3 Report
Reviewer Comments
This manuscript is about the feasibility study “if iCBCTs allow for reduced margin expansions accounting for inter-fractional motion while maintaining good target coverage.”
This manuscript provides a useful guideline for the reasonable margin for prostate radiotherapy using iCBCT image guidance which is especially useful to spare the OARs if the comparable CTV dose is reachable. However, this manuscript is still need to modify in order to be published in this journal. Please see the review comments as follows. My recommendation is to accept this manuscript with the minor revision.
Major Points:
1) Line 81, instead showed percentage such as 66Gy/33 fractions (71.4%), the reviewer suggested to change to patient number using the same prescription since this manuscript only has14 patients.
2) The authors mentioned several time in this manuscript to use Rapidplan to generate the treatment plan because of minimal human intervention. The reviewer suggested to add detail dose constraints for RapidPlan model rather than the new plan, since the new plan used the same RapidPlan model as the original plan.
3) Lines 213-214, the authors state that “Our study validated that 0-mm and 2-mm inter-fractional margins maintained sufficient CTV coverage and met dose constraints for the bladder and rectum”, but they should be careful not to generalize sufficient CTV coverage given that there was under-dosing for D100% in 32% of patients with 0 mm margins. The reviewer suggested to re-write this sentence.
4) The authors acknowledge that their small size is a limitation, especially because there are patients with (n=8) and without (n=6) endorectal balloons (ERB). When discussing the results, the authors necessarily differentiate between patients with and without ERB, but these small numbers do not convincingly justify that the results capture the overall impact of 0 mm, 2 mm, and/or 4 mm margins using iCBCT for post-prostatectomy patients. It would be helpful to include additional patients in the study cohort. This could be achieved one of two ways – (1) with recruitment of additional patients or (2) retrospective comparison between past plans and newly generated margin plans.
5) The authors suggest that consistency in prostate anatomy and planning dose distribution overcomes the limitation of sample size. However, the converse tends to be true, where post-prostatectomy plans tend to demonstrate the greatest variability in CTV and dose distribution (compared to prostate + seminal vesicles or prostate-only, for example) due to the larger area and extensive diversity in patient anatomy. So the reviewers suggested the author to include some discuss such as whether this study results are applied to all type of prostate bed patient or only apply to certain range of tumor volumes or tumor anatomy?
6) The reviewer also suggested to add some discuss of why only use isotropic PRV margin expansion? Why not add non-isotropic margin such as tighter margin as 0- or 2-mm margin in the superior and inferior direction and relatively larger margin such as 4-mm in the other directions?
Minor Points:
1) There is a figure citation discrepancy in lines 156-157, where the authors write “figure 3” and cite “4a-c.” Consistent figure references should also be used (i.e. “figure 4a-c” rather than “4a-c” alone).
1) There is another figure citation discrepancy in lines 166-167, where the authors write “figure 3” and cite “figure 4g-i.”
2) Text sizes (especially legends) should be increased for better readability in figure 2, figure 4, and figure 5.
3) Figure quality is very low for figure 3. Please increase the text sizes and re-add, so that the graphs are not blurry.
4) The authors sometimes refer to “endorectal balloons” and “rectal balloons.” There should be consistency.
5) Missing citation for paper referenced in lines 198-199.
6) The authors frequently use acronyms that are not always pre-defined in the text.
Author Response
This manuscript is about the feasibility study “if iCBCTs allow for reduced margin expansions accounting for inter-fractional motion while maintaining good target coverage.”
This manuscript provides a useful guideline for the reasonable margin for prostate radiotherapy using iCBCT image guidance which is especially useful to spare the OARs if the comparable CTV dose is reachable. However, this manuscript is still need to modify in order to be published in this journal. Please see the review comments as follows. My recommendation is to accept this manuscript with the minor revision.
Major Points:
- Line 81, instead showed percentage such as 66Gy/33 fractions (71.4%), the reviewer suggested to change to patient number using the same prescription since this manuscript only has14 patients.
Response: Thank you for the comment. The study focused on the dose delivery of each fraction based on daily iCBCT. The prescription scheme will not affect the study conclusion. We have further clarified the patient dose/fractionation scheme for the studied patients in the manuscript.
The most common dose/fractionation was 66 Gy in 33 fractions, which was delivered in 10 patients (71.4%). Other dose/fractionation schemes, including 52.5 Gy/20 fractions and 70.2 Gy/39 fraction, were delivered in 2 patients each (17.4%).
- The authors mentioned several time in this manuscript to use Rapidplan to generate the treatment plan because of minimal human intervention. The reviewer suggested to add detail dose constraints for RapidPlan model rather than the new plan, since the new plan used the same RapidPlan model as the original plan.
Response: The RapidPlan dose constraints were stated in the manuscript: “bladder V65% < 60%, bladder D0.03cc <108%, rectal wall D0.03 < 108%, CTV D95% > 95% of the prescription dose”
- Lines 213-214, the authors state that “Our study validated that 0-mm and 2-mm inter-fractional margins maintained sufficient CTV coverage and met dose constraints for the bladder and rectum”, but they should be careful not to generalize sufficient CTV coverage given that there was under-dosing for D100% in 32% of patients with 0 mm margins. The reviewer suggested to re-write this sentence.
Response: Thank you for bringing this to our attention. We have eliminated 0-mm margins from this sentence.
- The authors acknowledge that their small size is a limitation, especially because there are patients with (n=8) and without (n=6) endorectal balloons (ERB). When discussing the results, the authors necessarily differentiate between patients with and without ERB, but these small numbers do not convincingly justify that the results capture the overall impact of 0 mm, 2 mm, and/or 4 mm margins using iCBCT for post-prostatectomy patients. It would be helpful to include additional patients in the study cohort. This could be achieved one of two ways – (1) with recruitment of additional patients or (2) retrospective comparison between past plans and newly generated margin plans.
Response: Thank you for this important point. We recognize this is a limitation of our study. Although we use only 14 patients, 165 iCBCT were evaluated for this specific study. Due to the similarity of the prostate anatomy and planned dose distribution, we felt that our study data size of 165 is sufficient to draw the conclusion, given that this data size covered a large variation in bladder and rectum for prostate bed patients. We have indicated in the limitations section: “However, given the similar anatomy of prostate bed patients, a large variation of bladder and rectum volume and shape was accounted for, despite the small cohort of patients.”
- The authors suggest that consistency in prostate anatomy and planning dose distribution overcomes the limitation of sample size. However, the converse tends to be true, where post-prostatectomy plans tend to demonstrate the greatest variability in CTV and dose distribution (compared to prostate + seminal vesicles or prostate-only, for example) due to the larger area and extensive diversity in patient anatomy. So the reviewers suggested the author to include some discuss such as whether this study results are applied to all type of prostate bed patient or only apply to certain range of tumor volumes or tumor anatomy?
Response: Thank you for this important point. We have modified our discussion accordingly.
This study has several limitations. Although PTV margins were evaluated on 165 daily fractions of iCBCT, this only corresponded to 14 patients. However, given the similar anatomy of prostate bed patients, we believe we are accounting for bladder and rectum variation, despite the small cohort of patients. According to Yoon and colleagues, there was minimal deformation in CTV volume in patients receiving post-prostatectomy SBRT [11].In a follow-up study involving MR guidance published by Cao et al., there was slight variation in shape change of the CTV with dice similarity coefficient of 0.83 [25]. Volume changes reported in this study cannot be directly applied to our study as changes depend on the specific study group and treatment duration. As more recent trials have introduced shorter courses of post-prostatectomy radiotherapy, CTV volume changes are likely to be smaller than changes previously reported. Additionally, the development of prostate bed CTV guidelines has been defined by recurrence patterns established after landmark trials. Prostate bed CTV defined on the older version of CBCTs is associated with high user variation, and might not be accurate given poor image quality. As shown in Part I, iCBCT with improved image quality provides higher confidence in determining daily prostate bed CTV. We have confirmed no deformation of the prostate bed CTV in 165 datapoints for the 14 patients we studied. Nevertheless, this is a limitation of our study, which is that the conclusion applies to common anatomy without large size variations. We are expected to see higher dosimetric variation for those patients with potentially larger anatomy variations, which can then be discovered and monitored by high quality iCBCT images. The PTV margin for those scenarios should be subject to physician discretion. Additionally, we recognize the lack of evaluation of non-isotropic margins as a limitation of our study. [23] However, there may be heterogeneity in the field with application of isotropic or non-isotropic expansions. It is our clinical practice to use isotropic expansion.
- The reviewer also suggested to add some discuss of why only use isotropic PRV margin expansion? Why not add non-isotropic margin such as tighter margin as 0- or 2-mm margin in the superior and inferior direction and relatively larger margin such as 4-mm in the other directions?
Response: Thank you for this comment. We recognize this as a limitation of our study. Please see updated our limitations (yellow highlighted)
This study has several limitations. Although PTV margins were evaluated on 165 daily fractions of iCBCT, this only corresponded to 14 patients. However, given the similar anatomy of prostate bed patients, we believe we are accounting for bladder and rectum variation, despite the small cohort of patients. According to Yoon and colleagues, there was minimal deformation in CTV volume in patients receiving post-prostatectomy SBRT [11].In a follow-up study involving MR guidance published by Cao et al., there was slight variation in shape change of the CTV with dice similarity coefficient of 0.83 [25]. Volume changes reported in this study cannot be directly applied to our study as changes depend on the specific study group and treatment duration. As more recent trials have introduced shorter courses of post-prostatectomy radiotherapy, CTV volume changes are likely to be smaller than changes previously reported. Additionally, the development of prostate bed CTV guidelines has been defined by recurrence patterns established after landmark trials. Prostate bed CTV defined on the older version of CBCTs is associated with high user variation, and might not be accurate given poor image quality. As shown in Part I, iCBCT with improved image quality provides higher confidence in determining daily prostate bed CTV. We have confirmed no deformation of the prostate bed CTV in 165 datapoints for the 14 patients we studied. Nevertheless, this is a limitation of our study, which is that the conclusion applies to common anatomy without large size variations. We are expected to see higher dosimetric variation for those patients with potentially larger anatomy variations, which can then be discovered and monitored by high quality iCBCT images. The PTV margin for those scenarios should be subject to physician discretion. Additionally, we recognize the lack of evaluation of non-isotropic margins as a limitation of our study. [23] However, there may be heterogeneity in the field with application of isotropic or non-isotropic expansions. It is our clinical practice to use isotropic expansion.
Minor Points:
1) There is a figure citation discrepancy in lines 156-157, where the authors write “figure 3” and cite “4a-c.” Consistent figure references should also be used (i.e. “figure 4a-c” rather than “4a-c” alone).
Response: Thank you for this point. We have made this correction.
2) There is another figure citation discrepancy in lines 166-167, where the authors write “figure 3” and cite “figure 4g-i.”
Response: Thank you for this point. We have made the correction.
Figure 3 demonstrates V65% relative to daily bladder volume as a function of daily bladder volume ( Figure 3a-c) for uniform PTV expansions of 0-, 2-, and 4 mm. For each margin expansion, it is demonstrated that the bladder V65% < 60% is met in most cases as bladder volume approaches 100-150 cc. Figure 3a-c demonstrates that the V65 of the bladder decreases as the daily volume of the bladder increases. The bladder V65% < 60% constraint was not met in 4 (2%), 5 (3%), and 6 (4%) fractions for 0-, 2-, and 4 mm margins, respectively. The median bladder V65% was 17.6%, 18.6%, and 19.7% for 0-, 2-, and 4 mm margins, respectively. The interfraction median absolute change in bladder V65% was 4-4.8% for the different margin expansions. The maximum rectal wall (3mm) dose is demonstrated as a function of daily bladder volume for various target margins in Figure 3d-f. All fractions for the 0- and 2 mm margin plans resulted in a maximum rectal wall dose that fell under the 108% limit. In plans with a 4 mm PTV margin, three (3%) fractions resulted in maximum rectal wall doses that exceeded 108%, all in patients with an endorectal balloon. The median rectum D0.03cc were 102.2%, 103.7%, and 104.3% for 0-, 2-, and 4 mm in patients with an endorectal balloon, and 101.9%, 103.0%, and 103.9% for 0-, 2-, and 4 mm in patients without an endorectal balloon. The maximum bladder dose (D0.03 cc) is provided in Figure 3 as a function of normalized daily bladder volume (Figure 3g-i).
3) Text sizes (especially legends) should be increased for better readability in figure 2, figure 4, and figure 5.
Response: Legend font size has been adjusted to be same size as the font of text in the manuscript.
4) Figure quality is very low for figure 3. Please increase the text sizes and re-add, so that the graphs are not blurry.
Response: Thank you for this comment. Figure 3 has been adjusted.
5) The authors sometimes refer to “endorectal balloons” and “rectal balloons.” There should be consistency.
Response: We have adjusted to say endorectal balloon instead of rectal balloon.
6) Missing citation for paper referenced in lines 198-199.
Response: Thank you for this point. Part 1 of this study was submitted to Cancers with Part 2. It is also being reviewed simultaneously to Part 1. It can be included as a citation when published.
7) The authors frequently use acronyms that are not always pre-defined in the text.
Response: We have added an acronym list.
Acronym List
ANOVA: analysis of variance
CBCT: cone beam computed tomography
CTV: Clinical target volume
DVH: dose volume histogram
ERB: endorectal balloon
FDK: Feldkamp-Davis-Kress
FROGG: Faculty of Radiation Oncology Genito-Urinary Group
GU: genitourinary
HU: hounsfield Unit
iCBCT: iterative cone beam computed tomography
IGRT: image-guided radiotherapy
MRI: magnetic resonance imaging
OAR: organ-at-risk
PTV: planning target volume
RT:radiotherapy
SBRT: stereotactic body radiotherapy
VMAT: volumetric modulated arc therapy

Reviewer 4 Report
The authors investigated the dosimetrical effect of the PTV margin-reduction based on iCBCT in post-prostatectomy treatments in this retrospective study. The study is well designed and well written. Margin reduction is of particular interest in this patient population to reduce normal organs toxicity while maintaining the target coverage. One suggestion I have would be showing the results with the iCBCT image in addition or instead of the planning CT since the article is mainly focused on the IGRT advancement in iCBCT. It would provide more intuitive information to the readership on the image quality of the iCBCT. My other suggestion to the authors is to fix the full name of iCBCT in the abstract. It should not be "interactive cone-beam CT".
Author Response
The authors investigated the dosimetrical effect of the PTV margin-reduction based on iCBCT in post-prostatectomy treatments in this retrospective study. The study is well designed and well written. Margin reduction is of particular interest in this patient population to reduce normal organs toxicity while maintaining the target coverage. One suggestion I have would be showing the results with the iCBCT image in addition or instead of the planning CT since the article is mainly focused on the IGRT advancement in iCBCT. It would provide more intuitive information to the readership on the image quality of the iCBCT. My other suggestion to the authors is to fix the full name of iCBCT in the abstract. It should not be "interactive cone-beam CT".
Response: Thank you for this important comment. We certainly agree. Please refer to the paragraph below which is extracted from Part 1 of this manuscript series. We explain how iterative CBCT may provide enhanced image quality.
Figure 1 provides an example of image registration accuracy using iCBCT compared to the planning CT. The rectal wall and bladder interface and the prostate bed from the obturator internus musculature in patients can be clearly differentiated on the iCBCT with and without an endorectal balloon. This soft tissue contrast quality allows therapists to accurately align the patient without seeing a solid target in a post-prostatectomy setting with clear IGRT matching instructions.
Figure 1. Axial, coronal, and sagittal views demonstrating planning CT and iCBCT image guidance to identify rectal wall and bladder interface and space between bladder and obturator musculature.

Reviewer 5 Report
In this manuscript, the authors investigated the feasibility of margin reduction for prostate bed radiotherapy with advanced image guidance iCBCT. Overall, this is a very interesting topic with strong clinical relevance to clinical practice. The study is also unique in its approach which is different than the traditional margin estimation method. The manuscript in general well written, however, the quality could be improved in the following aspects:
- The authors need to clearly point out in the title and text that the findings of this study are related to fractionated radiotherapy treatment. In addition to all the factors evaluated in this study, margin selection also depends on the plan dose distribution. Considering the significant difference in dose gradient between fractionated and SBRT plans, the estimated margin in this study may not be directly translated to SBRT regimens.
- One major assumption of this study is minimal daily CTV shape change. Therefore, CTV is rigidly copied between planning dataset to daily CBCT. The authors referred to a publication (Yoon et al.) that reported an overall minor CTV volume change of 18 patients. However, 5 of 18 patients in their study exhibited significant under-coverage due to relatively larger volume changes which were also attributed to shape change. In a follow-up study involving MR guidance, the same group reported notable shape change of CTV (median DSC of 0.83 between planning and daily CTV). I think the assumption of this study may underestimate the deformable nature of the prostate bed and needs to be discussed as a limitation in the discussion section.
Specific comments:
1. Abstract line 24: The number of 32% is very confusing here, especially with the “slightly” under-doses. Recommend stating as CTV D100% was under-doses with 0-mm margin in 32% of the fractions.
2. Abstract line 25: CTV D95%>95% for 93-94% for all PTV expansions. This is not a complete sentence. Maybe: CTV CTV D95%>95% was met in 93-94% fractions for all PTV expansions. There are a few sentences like this in the entire text. Please revise them.
3. Abstract line 26: This sentence is very confusing. It reads like you are compared between 0, 2, 4 mm margins. After reading the entire text, I realize that 0.32% was from the comparison of all the margins with planning data. Also is this V95% or D95%? From M&M, seems like all Dxx parameters were analyzed for CTV.
4. Abstract: The result section did not capture sufficient data. For example, there is no comparison between the balloon and non-ballon patients, but your conclusion includes different margins for these two groups. I would recommend removing “intra-fractional” in the conclusion since this was not directly validated.
5. There are a lot of dosimetric parameters and constraints analyzed in this study, but they are reported inconsistently. For example, CTV D95%>95%, D100%>95% and D95%>90% were reported as dose constraints in different sections. Please review and keep them consistent.
6. Line 103: within 2% dose calculation accuracy.
7. Line 108: “may only be accounted by … prostate bed target definition”. This is confusing. If it is changed based on the target definition, one should re-contour the CTV on the iCBCT.
8. Figure 1. The isodose color bar for panels D-F may be wrong. There is a brown low-dose isodose line on the images, but not on the color bar.
9. Figure 3: Compared with Figures 2 and 4, it is really hard to interpret Figure 3. The manuscript did not provide any explanation or discussion other than simply reporting some values. It will be great to provide some discussion of this figure and what readers to learn from it.
10. Figure 4 and 5. It is great to see that the delivered doses to OARs were improved due to reduced margins while it is interesting that the differences in the rectum are more statistically significant than bladder. Any explanation? There are also notable fractions exceeding the dose constraints (dash line). Were they due to daily anatomy changes or because the initial planning parameters also exceed?
11. Discussion: line 198 “reduced bladder and maximum rectal dose”: From the results, I can see some reduced rectum volumetric doses, but the difference in bladder dosimetry is not statistically different.
12. Discussion: line 200: If D100%>95%, it implies all the rest parameters >95%. Therefore, clinically these parameters may have higher constraints or you don’t have to report here.
13. Line 214: Cleary some of the rectum/bladder constraints were not met based on Figure 4 and 5.
14. Line 228-241: This paragraph is very confusing. It reads like you try to infer the overall margin based on literature. But I would not call “study design conclude… ” with clear data support from this study. I would recommend rephrasing as “our study suggests that a margin of 2-3mm and 4-5mm might be sufficient in….”
Author Response
In this manuscript, the authors investigated the feasibility of margin reduction for prostate bed radiotherapy with advanced image guidance iCBCT. Overall, this is a very interesting topic with strong clinical relevance to clinical practice. The study is also unique in its approach which is different than the traditional margin estimation method. The manuscript in general well written, however, the quality could be improved in the following aspects:
- The authors need to clearly point out in the title and text that the findings of this study are related to fractionated radiotherapy treatment. In addition to all the factors evaluated in this study, margin selection also depends on the plan dose distribution. Considering the significant difference in dose gradient between fractionated and SBRT plans, the estimated margin in this study may not be directly translated to SBRT regimens.
Response: Thank you for this comment. We have mentioned fractionated radiotherapy in the title. We have made the following addition to the conclusions section.
An isotropic expansion of 2 mm and 4 mm may be considered for margin expansion with and without the endorectal balloon, respectively for fractionated radiotherapy. Future application for margin reduction needs to be further evaluated and considered with advent of shorter courses of post-prostatectomy radiation.
- One major assumption of this study is minimal daily CTV shape change. Therefore, CTV is rigidly copied between planning dataset to daily CBCT. The authors referred to a publication (Yoon et al.) that reported an overall minor CTV volume change of 18 patients. However, 5 of 18 patients in their study exhibited significant under-coverage due to relatively larger volume changes which were also attributed to shape change. In a follow-up study involving MR guidance, the same group reported notable shape change of CTV (median DSC of 0.83 between planning and daily CTV). I think the assumption of this study may underestimate the deformable nature of the prostate bed and needs to be discussed as a limitation in the discussion section.
Response: Thank you for this important comment. We have adjusted the following in the limitations section.
This study has several limitations. Although PTV margins were evaluated on 165 daily fractions of iCBCT, this only corresponded to 14 patients. However, given the similar anatomy of prostate bed patients, we believe we are accounting for bladder and rectum variation, despite the small cohort of patients. According to Yoon and colleagues, there was minimal deformation in CTV volume in patients receiving post-prostatectomy SBRT [11].In a follow-up study involving MR guidance published by Cao et al., there was slight variation in shape change of the CTV with dice similarity coefficient of 0.83 [25]. Volume changes reported in this study cannot be directly applied to our study as changes depend on the specific study group and treatment duration. As more recent trials have introduced shorter courses of post-prostatectomy radiotherapy, CTV volume changes are likely to be smaller than changes previously reported. Additionally, the development of prostate bed CTV guidelines has been defined by recurrence patterns established after landmark trials. Prostate bed CTV defined on the older version of CBCTs is associated with high user variation and might not be accurate given poor image quality. As shown in Part I, iCBCT with improved image quality provides higher confidence in determining daily prostate bed CTV. We have confirmed no deformation of the prostate bed CTV in 165 datapoints for the 14 patients we studied. Nevertheless, this is a limitation of our study, which is that the conclusion applies to common anatomy without large size variations. We are expected to see higher dosimetric variation for those patients with potentially larger anatomy variations, which can then be discovered and monitored by high quality iCBCT images. The PTV margin for those scenarios should be subject to physician discretion. Additionally, we recognize the lack of evaluation of non-isotropic margins as a limitation of our study. [23] However, there may be heterogeneity in the field with application of isotropic or non-isotropic expansions. It is our clinical practice to use isotropic expansion.
Specific comments:
- Abstract line 24: The number of 32% is very confusing here, especially with the “slightly” under-doses. Recommend stating as CTV D100% was under-doses with 0-mm margin in 32% of the fractions.
Response: We have made changes in the abstract to clarify this point.
CTV D100% was underdosed with 0-mm margin in 32% of fractions in comparison with 2-mm (6%) and 4-mm (6%) PTV margin (p < 0.001).
- Abstract line 25: CTV D95%>95% for 93-94% for all PTV expansions. This is not a complete sentence. Maybe: CTV CTV D95%>95% was met in 93-94% fractions for all PTV expansions. There are a few sentences like this in the entire text. Please revise them.
Response: Thank you for this comment. We have made corresponding changes in the abstract and the manuscript.
CTV D95% > 95% was met in 93-94% fractions for all PTV expansions.
- Abstract line 26: This sentence is very confusing. It reads like you are compared between 0, 2, 4 mm margins. After reading the entire text, I realize that 0.32% was from the comparison of all the margins with planning data. Also is this V95% or D95%? From M&M, seems like all Dxx parameters were analyzed for CTV.
Response: Thank you for this comment. We have evaluated D95% and this was corrected in the D95%.
There was no difference in absolute median change in CTV D95% (0.32%) for 0-, 2-, and 4-mm margins.
- Abstract: The result section did not capture sufficient data. For example, there is no comparison between the balloon and non-balloon patients, but your conclusion includes different margins for these two groups. I would recommend removing “intra-fractional” in the conclusion since this was not directly validated.
Response: Thank you for this comment. The following sentence was added to the results section in the abstract:
CTV D95% > 95% was achieved in more patients with an endorectal balloon than those without: 0 mm - 90/91 (99%) vs. 63/74 (85%); 2 mm - 90/91 (99%) vs. 65/75 (87%); 4 mm - 90/90 (100%) vs. 63/73 (86%).
- There are a lot of dosimetric parameters and constraints analyzed in this study, but they are reported inconsistently. For example, CTV D95%>95%, D100%>95% and D95%>90% were reported as dose constraints in different sections. Please review and keep them consistent.
Response: Thank you for this comment. The following adjustment has been made in the materials/methods section:
The following dose volume points were accessed for daily delivery using 3 different PTV margins: Bladder V65% < 60%, bladder D0.03cc <108%, rectal wall D0.03 < 108%, and CTV D90%, D95%, D98%, and D100% > 90% and > 95% were evaluated.
- Line 103: within 2% dose calculation accuracy.
Response: Thank you for this suggestion. We have modified as recommended.
Three PTV-margin VMAT plans were copied from the original CT of each patient to the corresponding daily iCBCTs, which have been calibrated and proved to be within 2% dose calculation accuracy [14].
- Line 108: “may only be accounted by … prostate bed target definition”. This is confusing. If it is changed based on the target definition, one should re-contour the CTV on the iCBCT.
Response: Thank you for this note. We have eliminated prostate bed target definition from the sentence.
For each iCBCT, prostate bed CTV was copied and approved by a radiation oncologist, and organs-at-risk were contoured. For every iCBCT, little to no deformation was appreciated and may only be accounted by intraobserver variation.
- Figure 1. The isodose color bar for panels D-F may be wrong. There is a brown low-dose isodose line on the images, but not on the color bar.
Response: Thank you for this comment. Figure 1 has been corrected.
- Figure 3: Compared with Figures 2 and 4, it is really hard to interpret Figure 3. The manuscript did not provide any explanation or discussion other than simply reporting some values. It will be great to provide some discussion of this figure and what readers to learn from it.
Response: We have added a sentence to help explain Figure 3.
Figure 3 demonstrates V65% relative to daily bladder volume as a function of daily bladder volume (Figure 3a-c) for uniform PTV expansions of 0-, 2-, and 4-mm. For each margin expansion, it is demonstrated that the bladder V65% < 60% is met in most cases as bladder volume approaches 100-150 cc. Figure 3a-c demonstrates the V65 of the bladder decreases as the daily volume of the bladder increases. The bladder V65% < 60% constraint was not met in 4 (2%), 5 (3%) and 6 (4%) fractions for 0-, 2-, and 4 mm margins, respectively.
The pertinent information regarding rectal wall doses (Figure 3d-f) is delineated in the results section.
- Figure 4 and 5. It is great to see that the delivered doses to OARs were improved due to reduced margins while it is interesting that the differences in the rectum are more statistically significant than bladder. Any explanation? There are also notable fractions exceeding the dose constraints (dash line). Were they due to daily anatomy changes or because the initial planning parameters also exceed?
Response: Thank you for this comment. They were added just for reviewers to have better visualization. There were not planning constraints used in this study. We have clarified this in our manuscript. We have added the following to the results discussion
Rectum DVH parameters (V40Gy, V65Gy, V70Gy, and V75Gy) are demonstrated in violin plots in Figure 4. The violin plots demonstrate the distribution of rectal dose in relation to rectum volume.
Figure 5: Violin plot for bladder DVH parameters for all patients. The parallel dash lines indicate the dose constraints. These DVH parameters were listed to show plan quality change. Dashed lines were commonly used dose constraints from literature or other practice guidelines.
- Discussion: line 198 “reduced bladder and maximum rectal dose”: From the results, I can see some reduced rectum volumetric doses, but the difference in bladder dosimetry is not statistically different.
Response: Thank you for this comment. We have made the following modification.
We demonstrated that a 0 mm PTV margin with an endorectal balloon and a 2 mm PTV margin without a endorectal balloon is feasible, accounting for inter-fractional uncertainty for post-prostatectomy RT, with adequate target coverage, reduced maximum rectum dose, and no change in bladder dose.
- Discussion: line 200: If D100%>95%, it implies all the rest parameters >95%. Therefore, clinically these parameters may have higher constraints or you don’t have to report here.
Response: We agree with the reviewer that if D100%>95% is met, all other parameters are met. However, we feel that it does not hurt to make them readily available to read in the paper, given that readers might use different dose constraints for their practice.
- Line 214: Cleary some of the rectum/bladder constraints were not met based on Figure 4 and 5.
Response: We have changed the sentence to say “with no significant change in meeting dose constraints for the bladder and rectum. Figure 4 and 5 show dvh points for bladder and rectum. They are not dose constraints. The dose constraints are listed in the manuscript “bladder V65% < 60%, bladder D0.03cc <108%, rectal wall D0.03 < 108%, CTV D95% > 95% of the prescription dose”. Figure 3 provided visual plots to show data points that these constraints were not met. They were all due to smaller bladder volume, thus we proposed “minimum bladder volume” concept in our Part I of the paper. An excerpt of Part 1, with explanation of the minimum bladder contour, is explained below.
A “minimum bladder” contour, the overlap between the original bladder contour and 15 mm anterior and superior expansion from prostate bed PTV, was confirmed to be effective in identifying cases that might fail bladder constraint of V65% <60%.
- Line 228-241: This paragraph is very confusing. It reads like you try to infer the overall margin based on literature. But I would not call “study design conclude…” with clear data support from this study. I would recommend rephrasing as “our study suggests that a margin of 2-3mm and 4-5mm might be sufficient in….”
Response: Thank you for this note. We have changed the last sentence to say “might be sufficient”
However, considering the abovementioned literature on intra-fractional motion, our study design concludes that a margin of 2-3 mm and 4-5 mm might be sufficient in ensuring CTV coverage with and without an endorectal balloon, respectively.

Round 2
Reviewer 2 Report
Thank you for modifying the discussion to address the concerns
Reviewer 5 Report
The authors have addressed all my concerns.